# Novel Signposts on the Road from Natural Sources to Pharmaceutical Applications: A Combinative Approach between LC-DAD-MS and Offline LC-NMR for the Biochemical Characterization of Two *Hypericum* Species *(H. montbretii* and *H. origanifolium*)

**DOI:** 10.3390/plants12030648

**Published:** 2023-02-01

**Authors:** Stefania Sut, Stefano Dall’Acqua, Gokhan Zengin, Ismail Senkardes, Abdullahi Ibrahim Uba, Abdelhakim Bouyahya, Abdurrahman Aktumsek

**Affiliations:** 1Department of Pharmaceutical and Pharmacological Sciences, University of Padova, Via Marzolo 5, 35131 Padova, Italy; 2Department of Biology, Science Faculty, Selcuk University, 42130 Konya, Turkey; 3Department of Pharmaceutical Botany, Pharmacy Faculty, Marmara University, 34722 Istanbul, Turkey; 4College of Science and Mathematics, Rowan University, Glassboro, NJ 08028, USA; 5Department of Biology, Faculty of Sciences, Mohammed V University in Rabat, Rabat 1014, Morocco

**Keywords:** *Hypericum*, hydroxycinnamic acids, flavonoids, natural enzyme inhibitors, bioactive agents

## Abstract

The members of the genus *Hypericum* have great potential to develop functional uses in nutraceutical and pharmaceutical applications. With this in mind, we aimed to determine the chemical profiling and biological properties of different extracts (ethyl acetate, methanol and water) from two *Hypericum* species *(H. montbretii* and *H. origanifolium*). We combined two approaches (LC-DAD-MS and LC-NMR) to identify and quantify chemical compounds of the extracts. Antioxidant properties (free radical quenching, reducing power and metal chelating) and enzyme inhibitory effects (cholinesterase, tyrosinase, amylase and glucosidase) were determined as biological properties. The tested extracts were rich in caffeic acid derivatives and flavonoids, and among them, 3-caffeoyl quinic acid and myricetin-3-*O*-rhamnoside were found to be the main compounds. The total phenolic and flavonoid levels were determined to be 50.97–134.99 mg GAE/g and 9.87–82.63 mg RE/g, respectively. With the exception of metal chelating, the methanol and water extracts showed stronger antioxidant properties than the ethyl acetate extracts. However, different results were obtained for each enzyme inhibition assay, and in general, the ethyl acetate extracts present more enzyme-inhibiting properties than the water or methanol extracts. Results from chemical and biological analyses were combined using multivariate analysis, which allowed establishing relationships between composition and observed effects of the *Hypericum* extracts based on the extraction solvents. To gain more insights between chemical compounds and enzyme-inhibiting effects, we performed molecular docking analysis. We observed favorable interactions between certain compounds and the tested enzymes during our analysis, confirming the data obtained from the multivariate approach. In conclusion, the obtained results may shed light on the road from natural sources to functional applications, and the tested *Hypericum* species may be considered potential raw materials, with promising chemical constituents and biological activities.

## 1. Introduction

Natural products refer to a broad category of chemicals derived from various organisms, including plants, fungi and animals. In the last century, however, the term has been commonly associated with plants or plant products. Plants have played a key role in traditional medicine since ancient times. Phytochemicals, particularly secondary metabolites, exhibit outstanding biological properties, ranging from antimicrobial to anticancer properties [1]. Because of their promising structural and pharmaceutical properties, these compounds are cornerstones as natural therapeutics in the development of novel drugs. Given the preceding fact, the research studies on non-investigated wild plants are increasing day by day in order to discovery new natural products [2,3,4,5]. The process of discovering and isolating natural bioactive compounds is complex and requires large amounts of solvents, plant material and time. At the same time, the repositioning of bioactive products is a hot research topic today, allowing the discovery of new effects of known compounds.

The genus *Hypericum* L. is one of the most popular genera in the various traditional medicine systems, including Anatolian folk medicine, and it is represented by almost 500 species in the world [6]. Among them, *H. perforatum* has been considered the most attractive, and many publications have focused on its chemical and biological properties [7]. Recently, some reviews considered ethnopharmacological uses in China [8], the use of Hypericum compounds as potential new drugs against COVID-19 [9], and the mechanism of hyperforin and hypericum extracts as anticancer agents through modulation of Inflammatory Signaling, ROS Generation and Proton Dynamics [10]. The long traditional use of the members of the *Hypericum* genus deal with multiple diseases. As an example, *Hypericum* spp. are commonly used to treat wound, depression, gastrointestinal and infection problems [6,8,11,12,13,14,15]. Moreover, the species have been utilized as tea and dietary supplements (as an oil). The members of the genus *Hypericum* contain biologically active compounds, including hypericin, hyperforin and flavonoids [16,17,18,19,20]. Although many articles have been published on the biopharmaceutical properties of the *Hypericum* species [21,22,23,24,25,26], information on the chemical and biological properties is still lacking for most of them. Among the species, *H. montbretii* and *H. origanifolium* are common in the eastern central and western regions of Anatolia and have been investigated in previous studies for their chemical content and some biological properties [17,27,28,29]. However, in the studies, the analytical methods for chemical characterization were generally simple colorimetric and liquid chromatographic techniques.

With this in mind, we aimed to develop a strategy of study of the plant material. Therefore, we created different extracts using solvent with increasing polarity, namely, ethyl acetate, methanol and water, from *H. montbretii* and *H. origanifolium*. With the aim to obtain accurate chemical fingerprinting, we used an approach of liquid chromatography and “offline NMR” using small preparative chromatography and NMR analysis. We combined the data with tandem mass spectrometry-based approaches. Related to this latter, we used liquid chromatography coupled with diode array and multiple stage mass spectrometry (LC-DAD-MS^n^) and high-resolution quadrupole time of flight mass spectrometry (HR-QTOF-MS). The orthogonality of the two spectral techniques, MS and NMR, offer the opportunity to have a comprehensive study of plant bioactive compounds. The novelty of our approach is dual, that is, the combination of the two spectral techniques and the choice of working on small scale; in fact, we used about 50–150 mg of dried extracts to perform the study. Information on the structure of bioactive compounds is recorded from the NMR of the fractions obtained by small-scale separations. The LC-MS approaches allowed obtaining qualitative and quantitative information on minor abundance constituents. Simultaneously, the acquisition of bioassay data (antioxidant and enzyme inhibitory properties) of the plant extract obtained with different solvents were used to correlate the chemical composition and spectral data. Finally, fine separations can be obtained from the most promising fractions using preparative TLC.

## 2. Results and Discussion

### 2.1. Phytochemical Profiling

The phenol and flavonoid content of the two plants extracted with the three solvents, namely ethyl acetate, methanol and water, were screened by colorimetric methods, and the results are presented in Table 1. We can note the richesse of the two plants by the phenolic compounds and flavonoids, with a fairly good variability between the two plants and between the three solvents used for the extraction. Indeed, the methanol and water extracts showed high content in both plants, with (131.11 ± 0.43 mg GAE/g and 134.99 ± 0.08 mg GAE/g, respectively) for *H. montbretii* and (106.92 ± 2.00 mg GAE/g and 93.34 ± 0.38 mg GAE/g, respectively) for *H. origanifolium* compared to the ethyl acetate extract. However, the ethyl acetate extract contained 50.97 ± 0.68 mg GAE/g and 71.58 ± 0.29 mg GAE/g for the two plants *H. montbretii* and *H. origanifolium*, respectively.

For the flavonoids, the differences are related both to the plant source and solvent. The methanolic extract shows a very high content of flavonoids compared to ethyl acetate in both plants, with values of 82.63 ± 0.4768 mg GAE/g for *H. origanifolium* and 68.57 ± 0.35 mg GAE/g for *H. montbretii*, followed by l aqueous extract, which recorded values of 63.59 ± 0.79 mg GAE/g and 62.30 ± 0.27 mg GAE/g, respectively, for *H. origanifolium* and *H. montbretii*.

These differences can be related to solvent polarities and compounds solubility, thus suggesting that the phenolic compounds in *H. origanifolium* and *H. montbretii* are better extracted with polar solvents. Indeed, these compounds are plant secondary metabolites with a wide range of pharmacological activities, such as anticancer, antiviral, anti-inflammatory, antidiabetic, antioxidant, etc. [30,31,32].

Previous studies, which have looked at other *Hypericum* species, have shown that the methanol extract of each plant contains a high amount of phenols and flavonoids [33,34]. Other phytochemical studies show that the aerial parts of *H. origanifolium* contain naphthodiantrones, flavonoids and xanthones [35].

Phenolics are a large and complex group of chemical constituents found in plants and are classic defense compounds to protect plants against herbivores and pathogens [36].

The differences in composition may be due to the geographical origins of the plant, environmental stimuli, climatic conditions and extraction methods, which affect the yield of secondary metabolites [37].

### 2.2. Strategy of Extract Fingerprinting Using Offline LC-NMR and LC-MS Based Methods

We previously used the analysis of methanol extract by NMR as a screening technique with other *Hypericum* species to drive the further investigations obtained by LC-MS [38]. To improve that approach, which suffers from the limited resolving power of NMR when directly applied to very complex mixtures as crude extracts, in this paper, we decided to perform fractionation of the extracts. For this reason, the ethyl acetate and methanol extracts, due to their lipophilic nature, have been subjected to a small silica gel column, using Si60 mesh fractionating in main fractions that will be used for NMR. The water extracts, due to the hydrophilic nature of the residue, have been separated using sephadex LH20. Results will be described for each specie. In parallel, two MS-based approaches, namely, analysis by LC-DAD-MS^n^ and LC-HR-QTOF, were obtained. Extracts will be finally tested on different bioassays. A schematic representation of the workflow is reported in Figure 1.

Annotation of main compounds was obtained by combining the information from 1D and 2D NMR data, allowing the identification of the main classes of constituents and, in some cases, the partial structure elucidation of some compounds. LC-DAD-MS^n^ and LC-HR-QTOF data were used to assess the complete qualitative and quantitative profiles of the different extracts.

### 2.3. Offline NMR Characterization of H. montbretii and H. origanifolium Fractions

Superimposition of the ^1^H spectra of *H. montbretii* ethyl acetate extracts fractions (Figure 2) allowed observing that the first fraction (Figure 2A) is mainly composed of lipids and chlorophylls, and the second (Figure 2B) presents signals that can be ascribed to specific secondary metabolites. In the C and D fraction, signals ascribable to phenolic glycosides and phenylpropanoids are evident and are highlighted by a square in Figure 2C,D. The enlargement of the aromatic part of the C and D ^1^H-NMR spectra are reported in Figure 3, and signals ascribable to myricetin (M), caffeic acid (C) and quercetin (Q) are indicated.

Assignments were confirmed by HSQC and HMBC data (See Table 2, and Appendix A). Considering the structures of the identified compounds (Figure 3), signals ascribable to positions H-6 and H-8 of flavonol nuclei appeared as broad singlets at δ 6.22 and 6.33, correlating in the HSQC-DEPT with carbon resonances at δ 98.4 and 93.4. HMBC correlations with carbon at δ 156.5 (C-9), 105.3 (C-10) and 163.5 (C-5/7) support the assignment. Furthermore, the singlet at δ 6.95 (δ 107.8) and the HMBC correlations (See Table 2) support this assignment to H-2′6′. A second series of signals, although less abundant, may support the presence of a 1,3,4 trisubstituted aromatic ring due to the presence of proton signals at δ 7.46, 7.45 and 7.21, assigned to positions 6′, 2′ and 5′ of quercetin due to the HSQC and HMBC correlation. Fraction C present intense signals, supporting the presence of myricetin and quercetin as major phenolic constituents (Figure 4). The presence of an anomeric proton signal (Table 2) and methylene and methyl group of rhamnose moiety, as shown in Figure 5, indicates that the most abundant compound is myricetin-3-*O*-rhamnopranoside. Fraction D, on the other hand, presents signals due to caffeic acid moiety and aliphatics signals, supporting the presence of quinic acid (K), as reported in Figure 5, thus revealing the presence of chlorogenic acid as the main constituent. Furthermore, minor signals in the aromatic region, namely, at δ_H_ 7.45–7.70 δ_C_ 120.0, can support the presence of anthraquinone derivatives.

Comparing the ethyl acetate fractions obtained from *H. origanifolium* (see Appendix A), we could observe that, as for the previous extract, the fraction A mostly contains lipids, while B, C and D contain the phenolic compounds. In the case of *H. origanifolium*, caffeic acid derivatives are missing and myricetin-3-*O*-rhamnoside resulted as the most abundant compound. Spectra of the *H. origanifolium* ethyl acetate fractions are reported as Appendix A.

The offline LC-NMR screening on the ethyl acetate fractions of the two hypericum species revealed that chlorogenic acid is efficiently extracted from *H. montbretii*, but not from *H. origanifolium*, by ethyl acetate. This behavior is unexpected, but can be related to the matrix effect, which did not allow efficient solubilization and extraction of this compound.

The methanol extracts of the two *Hypericum* were fractionated on silica gel, using and obtaining four fractions (A–D) for *H. montbretii* and five (A–E) for *H. origanifolium*; spectra are reported in the Appendix A. The main constituents that are revealed are also for this sample—myricetin, quercetin ad caffeic acid derivatives that are present in fractions B–D for *H. montbretii* and B-E for *H. origanifolium*. Some significant signals appear mainly in fraction C of *H. origanifolium*, and they are one sp^2^ methylene (δ_H_ 6.81; δ_C_ 137.4) and three oxymehtyne groups, one at δ_H_ 4.39, δ_C_ 66.2 one at δ_H_ 3.70, δ_C_ 69.8 and the last at δ_H_ 4.02, δ_C_ 66.5. Finally, an aliphatic CH_2_ is observed (δ_H_ 2.71–2.19, δ_C_ 30.2). All these signals are part of the same spin system, as observed in the COSY spectrum. Furthermore, the combination of HSQC-DEPT and HMBC allowed observing quaternary positions C-1 δ_C_ 128.2 and a carboxy function at δ_C_ 168.8. All the data indicate that the extract contains shikimic acid moiety. This moiety is also detectable in the *H. montbretii* methanol extracts, mostly in fraction B. In the same singlet ascribable to the methoxy group, (δ_H_ 3.70, δ_C_ 51.5) is evident, while it is not detected in other extracts. The comparison of the HSQC-DEPT and HMBC data allowed establishing the presence of quinic acid moiety (See Table 2) and showed a strong correlation from H-2/6 of the quinic acid (δ_H_ 2.34–2.05) with carboxyl function at δ_C_ 173.9, and the same HMBC correlation is observed from the methoxy group, thus indicating the presence of quinic acid methyl ester moiety. The spin system of the quinic acid moiety deduced by COSY and TOCSY appears to be multiple, but all can be ascribed to the quinic acid esterified in positions 3, 4 or 5 due to the unshielded chemical shift of some proton resonances (δ_H_ 5.30 and 4.23). Thus, from the NMR data, we can support in the methanol extract of *H. montbretii* the presence of a methoxylated derivative of quinic acid with ester linkage. In the methanol fraction of both the hypericum signals ascribable to hypericin or pseudohypericin are observed, namely, the aromatic proton H-2/5 (δ_H_ 6.50–6.70 δ_C_ 105–108) and H-9,12 (δ_H_ 7.45–7.70 δ_C_ 120–118), and also signals ascribable to methyl groups are observed (δ_H_ 2.95–3.00 δ_C_ 22.5).

Water fractions of the two *Hypericum* were fractionated using sephadex. Four fractions were collected, A–D, and the ^1^H-NMR spectra of *H. origanifolium* fractions (Figure 6) showed the differences clearly.

Fraction A presents signals due to sugars and polyphenols, fractions B and C present the clear signals of polyphenols and glycosides, while fraction D mainly contains lipids. The main compounds are the myricetin-3-*O*-rhamnoside, chlorogenic acid and shikimic acid detected in all fractions.

The water fraction of *H. montbretii* obtained with sephadex presents very similar chemical constituents as the *H. origanifolium* ones; spectra are reported in the Appendix A.

The assignment of the main position of the most abundant compounds has been performed by analyzing the 2D spectra obtained from the cleaner fractions, and the assignments are reported in Table 2. Spectra details with assigned positions are reported as figures in the Appendix A. Structures of the main compounds detected in the extract of *H. montbretii* are summarized in Figure 3. The NMR of the obtained fractions have evidenced the presence of flavonoid and caffeoylquinic derivatives as the main constituents in the plants. Thus, for further steps, preparative TLC was used to isolate the major constituents in the extracts used for the offline NMR.

### 2.4. Isolation of the Main Constituents from H. montbretii and H. origanifolium Fractions

Preparative TLC was selected as a profitable technique for the separation due to the limited amount of starting material and due to the small amount of the available extracts. After silica or sephadex separation, preparative TLC was used, and bands were scrapped and eluted with methanol. The isolated compounds were then characterized using NMR spectroscopy. The isolated compounds were chlorogenic acid, myricetin-3-*O*-rhamnopyranoside, quercetin-3-*O*-rhamnopyranoside and shikimic acid. Structures of the isolated compounds are reported in Figure 3.

### 2.5. LC-DAD-MS^n^ Characterization of H. montbretii and H. origanifolium

LC-DAD-MS^n^ was used to combine the detection of the UV active species and use the absorbance for quantitative purposes. Multiple stage mass spectrometry and UPLC-HR-QTOF were instead used for obtaining structural information on the eluted compounds. Four main classes of compounds, namely, quinic acid derivatives, flavonoids, phloroglucinols and anthraquinone derivative, were all detected, and the qualitative and quantitative data for the different plant extracts are summarized in Table 3 and Table 4.

The DAD chromatograms recorded at 330 nm are reported in Figure 7 and showed intense peaks at 11.9, 17.3 and 17.7 min for *H. montbretii* and at 11.9, 17.7, 18.9 min for *H*. *origanifolium*. The compounds have been identified as chlorogenic acid (3-caffeoyl quinic acid, 11.9 min), 7-methoxy-quinic caffeol ester (17.3 min), myricetin-3-*O*-rhamnoside (17.7 min) and quercetin-3-*O*-rhamnoside (18.9 min), in agreement with NMR data.

From a qualitative point of view, the main differences in the two species are related to the composition and extraction efficiency. *H. montbretii* extracts showed the larger amount of identified compounds. Phloroglucinol is different in the extracts of the two species, while caffeoylquinic acid derivatives are almost superimposable. *H. montbretii* extracts present as marker compounds flavonoid pentosides, a C-glucoside derivative of anthraquinone.

Considering the quantitative data, we can observe different behavior for the various classes of phytochemicals for the two species. Caffeoyl quinic acid derivatives are quite abundant in the extracts; in both the species, chlorogenic acid are the main derivatives. These two compounds are better extracted in methanol for both the species; we observed an efficient extraction of the different derivatives in water, indicating this solvent as appropriate for the extraction of these compounds.

The flavanones myricetin-3-*O*-rhamnoside and hyperoside are the most abundant flavonoid derivatives in both the species. This class of constituents were extracted with solvent, ethyl acetate, methanol and water, but for almost all the derivatives, methanol resulted as the best solvent.

The extracts of the two *Hypericum* species are not abundant in phloroglucinols, and in the analyzed samples, the composition is completely different in *H. origanifolium* and *H. montbretii.* Olympicon A is the most abundant in the *H. origanifolium* methanol extract (3 mg/g) compared to the other floroglucinol in both the plant species extracts. Hyperforin is detected in *H. origanifolium*, while hyperpolyfillirin in *H. montbretii.* Due to the different structure of these compounds, the most appropriated solvent is different for each of them.

Hypericin and pseudohypericin, belonging to antraquninone class, can be observed in the ethyl acetate fraction of *H. origanifolium*, while they are not detectable in the same solvent extract of *H. montbretii.* The methanol extract of both species exhibits only pseudohypericin, while water is able to extract both compounds in both species. The plant matrix and used solvent influence the extraction process, suggesting that specific protocols should be applied for anthraquinone derivatives. Considering the presented data for the two species, water appears to be the best solvent for the extraction of the three detected anthraquinone derivatives.

The proposed approach is valuable because it allows the identification of the main compounds and the confirmation of the structure of the most abundant due to isolation. Small flash chromatography on silica and gel permeation on Sephadex for the most hydrophilic solvents allow separating the most interfering compounds as lipids and polysaccharides and allow the improvement of NMR spectra. Application of 2D sequences help a lot for the compound identification. Small-scale preparative TLC allows obtaining compounds in sufficient purity and amount to confirm the structures of the most abundant derivatives, while comparison with standards helps to annotate further compounds. The accurate chemical analysis allowed establishing the differences between the two species, in particular, quercetin is more abundant in *H. montbretii*, and the 3′3′me6′oxo PIB derivative are most abundant in *H. montbretii*, while myricetin-3-*O*-rhamnoside is more abundant in *H. origanifolium*. Furthermore, we can observe that shikimic acid is quite abundant in both the plant extracts; in fact, we were able to isolate this compound, but it cannot be detected by LC-MS in the proposed methods. This shows the importance of using orthogonal approaches in natural product analysis to avoid losing information.

### 2.6. Antioxidant Properties

In our study, to study the antioxidant activity of our plant extracts, we used six methods: DPPH and the ABTS cation, FRAP, CUPRAC, phosphomolybdenum and metal chelating.

Table 5 illustrates the DPPH radical-scavenging activity of the different extracts. All the extracts tested showed a scavenging effect, while the extracts of *H. montbretii* presented a very important antioxidant activity compared to the extracts of *H. origanifolium*.

The aqueous and methanolic fractions of both plants showed increased activity of 340.05 ± 4.36 mg TE/g and 346.63 ± 2.86 mg TE/g for *H. montbretii* and 194.43 ± 2.67 mg TE/g, 266.36 ± 2.75 mg TE/g for *H. origanifolium*, respectively, while the ethyl acetate extract showed less activity for both plants. These results showed that the extracts contained a large amount of radical-scavenging phenolic compounds with proton-donating capacity.

For the ability of the extracts to scavenge the ABTS, cation was expressed in the table. The methanol fractions of *H. montbretii* and *H. origanifolium* had significant scavenging activity of 360.22 ± 0.56 mg TE/g and 402.73 ± 2.16 mg TE/g against ABTS, respectively. The aqueous fraction exhibited ABTS radical-scavenging activity. On the other hand, the antioxidant activity of the ethyl acetate extract is very weak for the two plants; this can be explained by the poor ability of this solvent to extract plant phenolics. By the CUPRAC method, the methanolic and aqueous extract of *H. montbretii* had high antioxidant activity (699.16 ± 2.52 mg TE/g, 637.33 ± 3.03 mg TE/g), respectively, compared to the methanolic and aqueous extract of *H. origanifolium*. With the FRAP method, a high absorbance indicates a high reducing power. The aqueous and methanolic fraction of the two plants showed very significant reducing activity.

In finding that the antioxidant activity of the extracts obtained with the different solvents is related to the amount of phenolic compounds. this correlation between the total phenolic compounds with the results of radical scavenging activity was observed in a previous study by Öztürk et al. [39]. These phenolic compounds are antioxidants with redox properties; the hydroxyl group helps them act as reducing agents, hydrogen donors and singlet oxygen quenchers [40]. Other results were observed in another study, including the trapping capacity of *Hypericum* has significant values (77.6% ± 0.5 for DPPH and 81.2% ± 0.4 for ABTS) and corresponds to the presence of a high quantity of phenolic compounds [41]. Another study finds *Hypericum* species to be good sources of natural antioxidants high in TPCs and major constituents [42].

For the phosphomolybdenum method, all the extracts of the two plants including the ethyl acetate extract showed antioxidant activities with similar values; it is a specific method. On the other hand, for the metal-chelating method, the ethyl acetate extract of *H. origanifolium* and *H. montbretii* showed very significant effects compared to other extracts, with values of 24.27 ± 0.93 mg EDTAE/g and 22.61 ± 0.37 mg EDTAE/g, respectively; it is too specific a method, whose ethyl acetate extracts contain molecules that have the ability to react with metals. In particular the presence of unsaturated fatty acid can at least in part explain this result. For example, the phosphomolybdenum activity of lipid extract of *Sorghum* seeds ranged from 0.13 to 0.21 µmol VEEAC (equivalent of vitamin E)/g in another study performed by Hadbaoui et al. [43]. In addition, Benalia et al. [44] reported that the total antioxidant abilities of pumpkin seed oils varied from 18.88 to 56.30 mg/mL (EC_50_ values). Several other studies, also using the phosphomolybdenum test, show that *Hypericum* had considerable antioxidant activities [45,46,47].

Indeed, the antioxidant activity depends on the interactions in the reaction media between the substrate(s) (radicals) and the active molecule(s) that trap them [48].

The effectiveness of the antioxidants can be attributed to the high amount of the main constituents, mostly the phenolics, and also to the presence of other constituents in small amounts or to the synergy between them.

### 2.7. In Vitro Antidiabetic Activity

α-amylase and α-glucosidase are the two key enzymes that break down complex sugar into simple sugar at the intestinal tract level. These degradations result in simple products, in particular glucose, which will be absorbed, and consequently, there will be an increase in blood sugar. One of the therapeutic approaches to improve diabetes is to lower postprandial blood sugar by inhibiting carbohydrate-hydrolyzing enzymes [49]. From this context, the inhibition of these enzymes at the intestinal level will block the degradation of complex sugar towards simple sugar, and consequently, it will contribute to the reduction of the consent of blood glucose. In our work, we studied the test of inhibition of these enzymes by the extracts of our plants. The extracts of our plants showed inhibitory effects on the two enzymes tested, as presented in Table 6. The results revealed that these extracts inhibit the activity of α-amylase and α-glucosidase.

These inhibitors may delay the absorption of dietary carbohydrates in the small intestine and reduce postprandial hyperglycemia, which may be a useful mechanism in the preparation of antidiabetic drugs [50].

Indeed, the ethyl acetate fraction of *H. montbretii* and *H. origanifolium* showed strong inhibitory capacity against α-Amylase (0.61 ± 0.02 and 0.63 ± 0.01 mg ACAE/g respectively). In addition, the methanol fraction showed moderate inhibition of α-glucosidase (1.14 ± 0.02 mg ACAE/g for *H. montbretii* and 1.09 ± 0.03 for *H. origanifolium*) and α-Amylase (0.52 ± 0.01 mg ACAE/g for *H. montbretii* and 0.49 ± 0.01 mg ACAE/g for *H. origanifolium*). However, only the aqueous extracts of *H. origanifolium* present an enzymatic activity against α-glucosidase with a value of 1.10 ± 0.02 mg ACAE/g.

These results are consistent with other studies that considered another species of *Hypericum*. Their study revealed that the methanolic extract of the whole plant exhibited α-glucosidase inhibitory activity, which increased with increasing concentration [51,52,53].

### 2.8. Cholinesterase Activity

Acetylcholinesterase (AChE) and butyrylcholinesterase (BChE) are enzymes specific to nervous tissues and neuromuscular junctions. They rapidly hydrolyze acetylcholine (neurotransmitter) into inert choline and acetate. AChE is involved in the process of nervous transmission, and consequently, a strong expression or an abundant catalysis creates disturbances at the neuron level, among the strategies used for neuroprotection is the blocking of AChE.

In this context we use our extracts as an inhibitor of these two enzymes (AChE and BChE). The results are indicated in Table 6.

The study of anti-AChE and anti-BChE activities of our studied extracts showed that these extracts are able to inhibit AChE and BChE. Indeed, the results showed that the ethyl acetate extract is more active against BChE, with a value of (6.61 ± 0.08 mg GALAE/g for *H. montbretii* and 4.94 ± 0.15 mg GALAE/g for *H. origanifolium*). Some paper reported that fatty acid can exert some inhibitory activities on these enzymes, and this can explain the activity observed [54,55].

In addition, the methanolic and aqueous extracts of the two plants are selectively active only against AChE, with values of 2.17 ± 2.12 mg GALAE/g and 1.55 ± 0.11 mg GALAE/g for the methanolic and aqueous extracts of *H. montbretii*, respectively, and of 2.17 ± 0.06 and 2.21 ± 0.04 for the two extracts of *H. origanifolium*. This study showed that most of the extracts are very significant in terms of AChE inhibitory power compared to BChE.

The therapeutic action of cholinesterase inhibitors is, therefore, essentially due to the inhibition of acetylcholinesterase at the central level.

According to other studies, it is noted that *Hypericum* has the ability to block and inhibit both types of enzyme [53,56,57,58,59,60,61]; in addition, other plants have this ability [62,63,64,65]. Finally, it is a major therapeutic strategy for neuroprotection.

### 2.9. Tyrosinase Activity

Tyrosinase is a key enzyme involved in skin cell aging, and its inhibition is an important strategy to delay skin aging. Tyrosinase catalyzes the first two common steps of melanogenesis and thus appears to be the limiting enzyme [66]. Its absence or mutations of its gene lead to a decrease or even a cessation of pigmentation. Mutation of his gene has been found to be associated with oculocutaneous albinism type I [67].

In our study we tested all the extracts against tyrosinase, and we demonstrated that all the extracts are active against with a variability that depends on both the plant studied and the solvent used; the results are presented in the Table 6. Ethyl acetate and methanol extracts from both plants showed very strong tyrosinase inhibitory activity, with values between 69.66 ± 0.47 mg KAE/g and 57.47 ± 0.98 mg KAE/g. On the other hand, the aqueous extract had presented a moderate activity against tyrosinase, with values of 34.47 ± 1.08 mg KAE/g for *H. montbretii* and 16.63 ± 1.46 mg KAE/g for *H. origanifolium*.

A great interest is focused on natural compounds capable of inhibiting the activity of tyrosinase, for which there is an increasing demand in the fields of cosmetic and pharmaceutical applications. In the literature, several studies which have shown that the members of the genus *Hypericum* exhibited inhibitory effects on tyrosinase [68,69,70,71]. To provide a structure-ability relationship, as can be seen in Table 3 and Table 4, some identified compounds in the tested extracts have been reported to be potent anti-tyrosinase inhibitors. For example, in a previous study by Lou et al. [72], myricetin-3-*O*-rhamnoside exhibited a good anti-tyrosinase inhibitor ability, with a lower IC_50_ value among some isolated compounds. In addition, the derivatives of caffeoylquinic acids, including 3-caffeoylquinic acid, displayed a significant tyrosinase inhibitory effect [73]. In another study by Park et al. [74], quercetin-*O*-rhamnoside was isolated, and it was tested on tyrosinase. The authors suggested that the compound could be useful in treating skin disorders. In this sense, the tested *Hypericum* species could be considered as sources of natural anti-tyrosinase agents in the preparation of effective cosmeceuticals.

### 2.10. Molecular Docking

To understand the interaction of the bioactive compounds with the target enzymes, molecular docking was performed. The binding energy (docking) score of each ligand against each target enzyme is displayed in Figure 8. All the study ligands showed potential binding to the 5 enzymes, with some of the compounds displaying a preference for AChE and BChE, amylase and glucosidase. Therefore, the detailed analysis of protein-ligand interactions was analyzed for some selected complexes. Quercetin 3-*O*-rhamnoside was predicted to have strong binding potential to both AChE and BChE, and it also bound to amylase, tyrosinase and glucosidase with high affinity. Quercetin 3-*O*-rhamnoside bound to AChE and BChE in different orientations and formed multiple H-bonds bonds, several van der Waals interactions, and a couple of hydrophobic interactions with amino acid residues lining the catalytic channel of AChE (Figure 9A), with additional π-π stacked interactions in the case of BChE (Figure 9B).

In the case of tyrosinase, with a relatively narrow pocket, myricetin 3-*O*-rhamnoside was accommodated via a couple of H-bonds, a π-π stacked interaction, a few Van der Waals interactions deep inside the tunnel, as well as a π-anion and π-cation interactions near the entrance to the pocket (Figure 9C). Interestingly, quercetin 3-*O*-galactoside (hyperoside) was completely buried in the active site of amylase, forming H-bonds, van der Waals interactions, π-π stacked interactions and a hydrophobic interaction deep inside (Figure 9D). Likewise, 3-Caffeoylquinic acid occupied the cavity of glucosidase via multiple H-bonds, a few Van der Waals interactions and a hydrophobic (Figure 9E). Together, these interactions may be responsible for the observed biological activity of the bioactive compounds on these target enzymes.

### 2.11. Multivariate Analysis

Due to the large amount of experimental data, it is difficult to establish relationships, but we wanted to study if there is any relation between the solvent used for the extraction and the results of the bioassays, and we also wanted to establish any relation, if present, between chemical composition and bioactivity. Figure 10 represent a PLS-DA obtained considering as X variables the chemical constituents in each extract and as Y the results of the different bioassays test. As we can observe in Figure 10 and in the loading scatter plot Figure 11, the results related to inhibitory assays are mostly occupying the −x + y part of the plot and are correlated mostly with the more lipophilic extracts of both the hypericum species. Considering the compounds, high correlation with enzyme inhibitory activity appears to be with quercetin-3-*O*-rhamnoside, hyperforin, 3′3′me6′oxo-PIB-derivative, geranylpholoroisobutylphenone and biapigenin, indicating that multiple compounds can act. This is expected since in this first elaboration, we considered all the enzymatic activities together, and we can expect that specific compounds can be related to each enzymatic activity. The biplot shown in Figure 12 gives a general overview of the graph.

The assays related to antioxidant activity are, on the other hand, mostly concentrated in the +x-y part of the plot, and the compounds mostly related to these activities appear to be quercetin and caffeoyl derivatives, as we can expect from literature data.

The loading scatter plot suggests that the compounds most significantly involved in the antioxidant activity of the analyzed extracts are the caffeoyl quinic acid, and the most abundant quercetin derivatives, namely, the hyperoside and the 3-*O*-rhamnoside. This result is obviously related to the specific structural moieties of the compounds, as well as to the amount of each compound in each extract. We, in fact, should always consider that in studying the plant extract activities, we are evaluating the effects of complex mixtures of compounds that act based on their chemical moieties, as well as due to their abundance in the tested extract.

As we have observed, the methanol and water extracts are rich in chlorogenic acid, myricetin and quercetin derivatives, and these compounds, due to their structure presenting phenolic groups, can act easily as antioxidants, as well as present significant metal-chelating properties. We can observe that the phloroglucinols are more correlated to the enzyme inhibitory effects, and we should consider this result as a consequence of their specific chemical structure that can probably help interactions with different sites of the enzymes, but their relatively minor role in these specific extracts can also be related to their low abundance compared with the phenolic derivatives.

## 3. Materials and Methods

### 3.1. Plant Materials and Extraction

The aerial parts of the plants (*H. montbretii*: Taskopru, between Beykoy and Bozarmut, 1375 m; *H. origanifolium*: Hanonu, between Yenikoy and Yilanli, 531 m) were collected in Kastamonu of Turkey in the summer season of 2020. The plant was identified by one botanist co-author (Dr. Ismail Senkardes, Marmara University). Voucher specimens were deposited in the herbarium at Marmara University (Voucher Numbers: MARE-18374 and MARE-19844, respectively).

In the preparation of plant extracts, we used three solvents (ethyl acetate, methanol and water) to extract compounds with different polarities. Maceration was selected for ethyl acetate and methanol extracts, and for this purpose, plant materials (10 g) were stirred with the 200 mL of methanol for 24 h at room temperature. After that, the mixtures were filtered using Whatman filter paper, and the solvents were removed using a rotary-evaporator. Regarding the water extract, the extract was prepared as a traditional infusion, and the plant materials (10 g) were kept in the boiled water (200 mL) for 15 min. Then, the mixture was filtered and lyophilized for 48 h. All extracts were stored at 4 °C until analysis.

The extraction yields were calculated based on the formula yield (%) = 100 × (W1/W2), where W1 is the mass of the crude extract (g) and W2 is the mass of the initial material (g) [75].

### 3.2. Profile of Bioactive Compounds

Folin–Ciocalteu and AlCl_3_ assays, respectively, were utilized to determine the total phenolic and flavonoid contents [76]. For respective assays, results were expressed as gallic acid equivalents (mg GAEs/g extract) and rutin equivalents (mg REs/g extract).

### 3.3. LC-DAD-MS^n^ and LC-QTOF Analysis of Hypericum montbretii and Hypericum Origanifolium Extracts

For the chemical characterization of the extracts, an Agilent 1260 system was used, coupled with a 1260-diode array (DAD) detector and an ion trap Varian MS 500. An Eclipse XDB C18 3 × 150 mm 3.5 µm was used, and the mobile phases were water (1% formic acid) (A), acetonitrile (B) and methanol (C). The elution gradient was as follows: 95:5:0% (A:B:C) from 0 min to 0.5; 85:15:0% (A:B:C) at 10 min; 60:30:10% (A:B:C) at 15 min; 20:70:10% (A:B:C) at 20 min; 0:90:10% (A:B:C) at 25 min till 30 min; and 5 min of re-equilibration time. The flow was 0.3 mL/min and injection volume 5 µL. Each extract was exactly weighted (4 mg) and dissolved in 0.5 mL of DMSO. The solution was sonicated for 10 min and centrifuged for 5 min, and the liquid was used for analysis. With the diode array, chromatograms were acquired in the range 200–600 nm and traces recorded at 254 nm, 280 nm, 330 nm and 590 nm. For identification of each peak, UV spectra were acquired. A mass spectrometer was use with an Electrospray (ESI) ion source, and mass spectra were acquired in negative ion mode in a mass range between 100–1200 *m/z*. Ion trap collected data in TDDS mode allowed multiple reaction monitoring with multistage fragmentation, allowing the identification of secondary metabolites based on comparison with the reference standard and literature. Mass spectrometer parameters were the following: needle voltage 4500 volts, nebulizer gas pressure 25 psi, drying gas pressure 15 psi, drying gas temperature 260 °C, spray chamber temperature 50 °C, capillary voltage 80 volts and RF loading 80%. For quantification, chlorogenic acid, quercetin-3-*O*-glucoside, quercetin-3-*O*-galactoside (hyperoside), hyperforin and hypericin were used. Standard solutions were prepared in methanol: water (50:50) for chlorogenic acid, methanol for quercetin-3-*O*-glucoside, quercetin-3-*O*-galactoside and hyperforin and methanol: DMSO (50:50) for hypericin, respectively. Standard solutions were prepared at four different concentrations in a range of 50–1 µg/mL, and calibration curves were calculated. For quantitative purpose, metabolites were grouped in phloroglucinols derivatives, anthraquinone derivatives, quinic acid derivatives and flavonoids. For the LC-QTOF analysis, a Waters Acquity UPLC system coupled to a Waters Xevo G2 QTOF mass spectrometric (MS) detector. As stationary phase, an Agilent Zorbax Eclipse Plus C18 (2.1 × 50 mm, 1.8 µm) column was used, and column temperature was maintained at 40 °C. A mixture of water + 1% formic acid (A) and methanol + 1% formic acid (B) was used as the mobile phase. The elution gradient was as follows: 0–1 min, 98% A; 11 min, 15% A; 16 min, 0% A; 20 min, 0% A; 21 min, 98% A; 24 min, 98% A. Flow rate was 0.3 mL/min, and the injection volume was 2 μL. MS data were acquired in negative ionization mode (ESI-) in the mass range 50–2000 Da. The sampling cone voltage was adjusted at 40 V, the source offset at 80 V. The capillary voltage was adjusted to 3.5 KV. The nebulizer gas used was N_2_ at a flow rate of 800 L/h. The desolvation temperature was 450 °C. The mass accuracy and reproducibility were maintained by infusing lock mass (leucine–enkephalin, [M−H]^−^ = 554.2620 *m/z*) through Lockspray at a flow rate of 20 μL/min. The m/z value of all acquired spectra was automatically corrected during acquisition based on lock mass. A MS^e^ experiment was simultaneously performed to collect structural information, setting the collision energy to 30 V.

### 3.4. Fractionation of Extracts, Offline NMR and Purification of Compounds

A total of 1.5 g of silica gel 60 mesh was loaded in a plastic cartridge for flash chromatography and packed; one cartridge each was packed for ethyl acetate and methanol extract. Ethyl acetate extracts of *H. montbretii* (67 mg) and *H*. *origanifolium* (69 mg) were used. The cartridge was loaded with the ethyl acetate extract solubilized in 0.2 mL of mobile phase and eluted with toluene: methanol in ratio 10:3 (50 mL). The stationary phase was then washed using 10 mL of methanol to elute more polar compounds. Next, 30 fractions of 2 mL were collected and pooled on the bases of the TLC behavior in four groups named A–D. Fractions were dried under vacuum. The NMR of the fraction were recorded in deuterated chloroform, and spectra are included in the Appendix A.

The same procedure was performed for methanol extract of *H. montbretii* (120 mg) and *H. origanifolium* (130 mg) with a different mobile phase, namely, dichloromethane: methanol 3:1. Then, 50 mL of mobile phase was used and followed by 10 mL of methanol for eluting all compounds. Fractions were grouped in four groups (A–D) for *H. montbretii*, while in five (A–E) for *H. origanifolium* due to the chromatographic behavior in TLC. The NMR of all the fractions were recorded in deuterated methanol, and spectra are included in the Appendix A.

For the analysis of water fractions, cartridges were filled with Sephadex LH20 (4.0 g), and for the elution, methanol was used (40 mL), and then 5 mL of acetone for final washing. Fractions were pooled in 4 groups based on the TLC behavior (A–D). The NMR of all the fractions were recorded in deuterated methanol, and spectra are included in the Appendix A.

From the fractions after the NMR spectra acquisition, we isolated the main constituents using preparative TLC. A Camag Lynomat 5 was used to charge the plates (Silica gel plates 20 × 20). Eluents used for separating compounds were mixtures of n-Buthanol:Acetic acid:Water (20:5:1). After plate development, spots related to the main compounds were detected by UV (254 nm) and scraped from the plate. Silica was washed with methanol, and liquid was filtered and dried under vacuum. Residues were dissolved in deuterated methanol and used for structure elucidation. From the two Hypericum, the following compounds were isolated: myricetin-3-*O*-rhamnopyranoside, chlorogenic acid and quercetin-3-*O*-rhamnopyranoside. Shikimic acid structures of the compounds were deduced from 1D, 2D NMR experiments and finally compared with reference standards available in the laboratory.

### 3.5. Determination of Antioxidant and Enzyme Inhibitory Effects

The antioxidant and enzyme inhibitory activity of the extracts was determined according to previously described methods [77,78]. DPPH and ABTS radical scavenging activity, cupric ion reducing antioxidant capacity (CUPRAC) and ferric ion reducing antioxidant power (FRAP) were expressed as mg Trolox equivalents (TE)/g extract. The metal-chelating ability (MCA) was reported as mg EDTA equivalents (EDTAE)/g extract, whereas the total antioxidant activity (phosphomolybdenum assay, PBD) was expressed as mmol TE/g extract. AChE and BChE inhibitory activities were given as mg galantamine equivalents (GALAE)/g extract; tyrosinase inhibitory activity was expressed as mg kojic acid equivalents (KAE)/g extract; and amylase and glucosidase inhibitory activities were presented as mmol acarbose equivalents (ACAE)/g extract.

### 3.6. Molecular Modeling

To gain insights into the interaction of the compounds from the tested extracts, a molecular docking simulation was carried out. The target enzymes’ crystal structures were downloaded from the protein data bank (PDB) (https://www.rcsb.org/ (accessed on 1 April 2022)) with the following IDs: human AChE (PDB ID: 6O52) [79] and BChE (PDB ID: 6EQP) [80], human pancreatic alpha-amylase (PDB ID: 1B2Y) [81]. However, the crystal structures of human tyrosinase and glucosidase are not available; therefore, *Priestia megaterium* tyrosinase (PDB ID: 6QXD) [82] and *Mus musculus* alpha-glucosidase (PDB ID: 7KBJ) [83] were retrieved and used as templates to build their human models using the respective human sequences and UniProt entries P14679 and P0DUB6. The detailed procedure of the model building was described previously in [84]. The pKa of titratable residues in each protein was predicted using the “Playmolecule ProteinPrepare” module [85] and was then used to prepare the proteins at a physiological pH of 7.4. The ligand 3D structures were retrieved from the PubChem database (https://pubchem.ncbi.nlm.nih.gov/ (accessed on 1 April 2022)) and optimized using Frog2 [86]. Docking grid files were generated based on the size of the active site of each enzyme and the binding (x,y,z) coordinates of the respective cocrystal ligand using AutoDockTools 1.5.6, followed by docking using AutoDock 4.2.6 (https://autodock.scripts.edu (accessed on 1 April 2022)) [87]. The details of the docking were described previously in [88,89,90,91]. The binding energy (docking) score of each ligand (pose) was estimated, and protein–ligand interactions were examined using Biovia Discovery Studio Visualizer (Dassault Systèmes Biovia Software Inc., 2012).

### 3.7. Statistical Analysis

Data are presented as mean ± standard deviation of the number (*n* = 3) of replicates. One-way analysis of variance with Tukey’s post-hoc test was conducted; *p* < 0.05 was considered statistically significant. The statistical evaluation was performed using GraphPad version 9.0. For the generation of the multivariate data analysis plot, SIMCA 12 was used. Quantitative data obtained from the LC measurements were used to generate a table containing all the extracts, all the quantified compounds and all the results of the bioassays. The matrix was loaded in SIMCA 12 and pareto scaled. Data initially were used to obtain a PCA, then a PLS-DA was used, assigning as Y variables all the data from the bioassays and X data all the quantified compounds.

## 4. Conclusions

The current work examined the chemical characterization and biological properties of different extracts from two *Hypericum* species, namely, *H. montbretii* and *H. origanifolium*. We used the combination of LC-MS-DAD and offline LC-NMR methods to detect chemical compounds in the tested extracts, and this is the first application on the members of the genus *Hypericum*. In the chemical profiles, quinic acids derivatives and flavonoids were recorded as the predominant groups. The biological properties of the tested extracts depended on the extraction solvents used, and generally, the methanol and water extracts were more active when compared to ethyl acetate. This fact was also supported by multivariate analysis, which provided a separation based on the extraction’s solvents for the tested *Hypericum* species. Molecular docking analysis also showed a connection between chemical compounds and the tested enzymes, and some compounds had a strong binding capacity on the enzymes. From this, it can be concluded that our findings could provide valuable contributions on the natural product area, and the *Hypericum* species could be considered valuable candidates for functional applications in nutraceutical, pharmaceutical and cosmeceutical industries. Nonetheless, we advocate for more research into the *Hypericum* species, particularly with regard to isolated compounds and their biological and toxicological properties.

## Figures and Tables

**Figure 1 plants-12-00648-f001:**
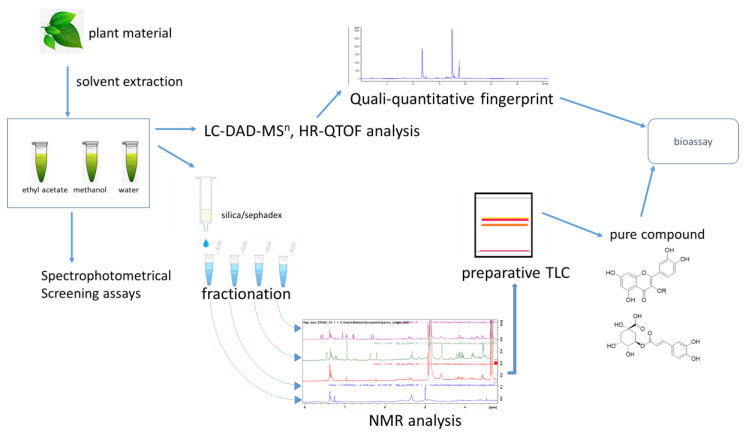
Workflow of the different measurements described in the paper.

**Figure 2 plants-12-00648-f002:**
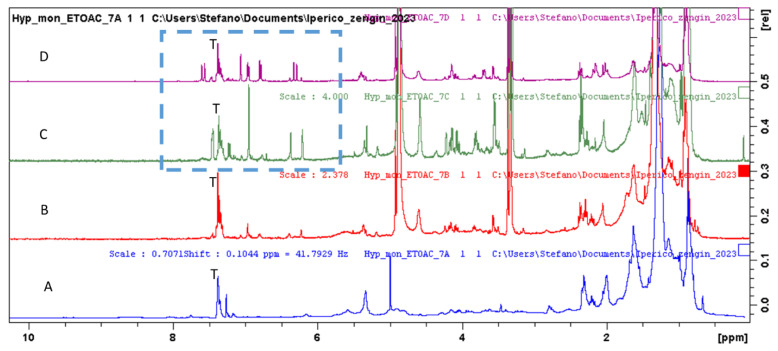
Superimposition of the ^1^H-NMR spectra of *Hypericum montbretii* ethyl acetate fractions (**A**–**D**). Blue square indicate the signals of phenolic compounds (T indicates signals due to toluene from the chromatographic fractionation process).

**Figure 3 plants-12-00648-f003:**
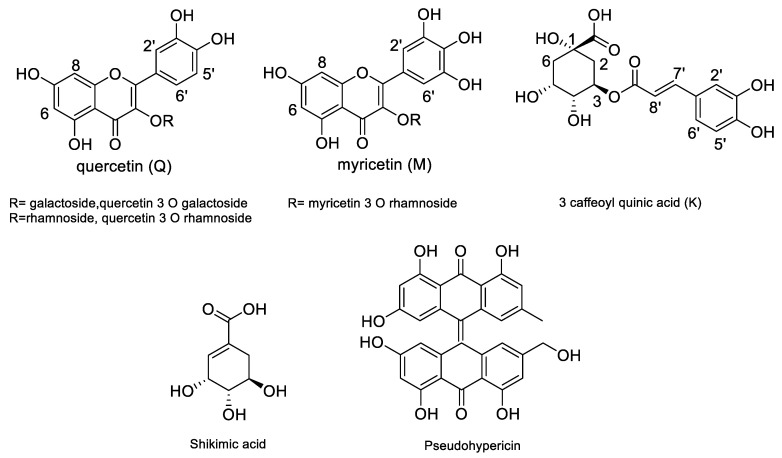
Structures of main compounds detected in the *H. montbretii* and *origanifolium*.

**Figure 4 plants-12-00648-f004:**
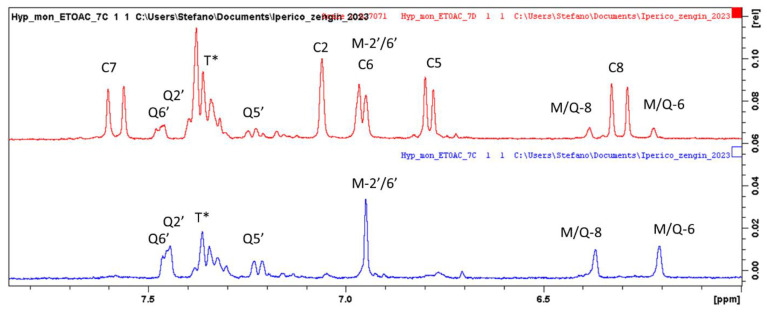
Enlarged part of the ^1^H-NMR spectra of *Hypericum montbretii* ethyl acetate fractions (C in red and D in blue). For assignments, see text.

**Figure 5 plants-12-00648-f005:**
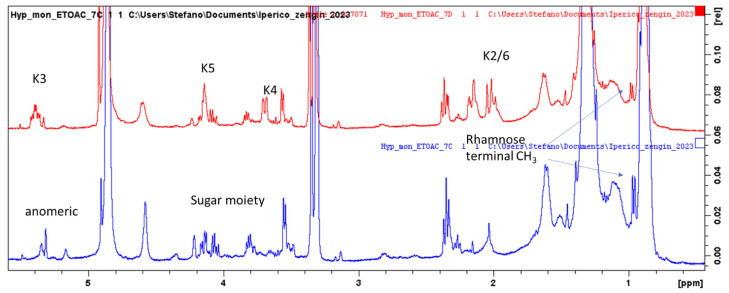
Superimposition of the ^1^H-NMR spectra of *Hypericum montbretii* ethyl acetate fractions. C (blue spectrum) and D (red spectrum) signals ascribable to quinic acid moieties (K) and sugar residue are indicated.

**Figure 6 plants-12-00648-f006:**
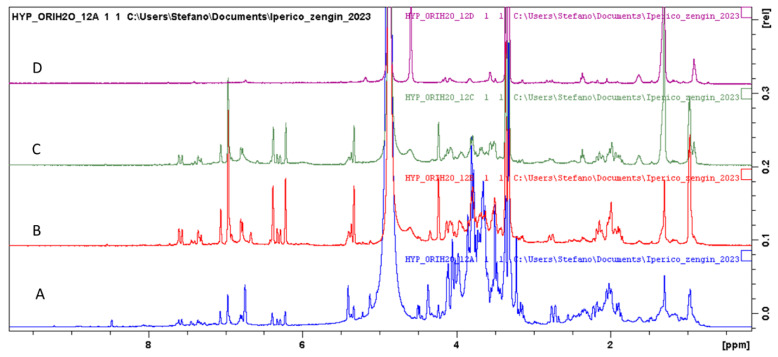
H-NMR of *H. origanifolium* water fraction obtained from sephadex.

**Figure 7 plants-12-00648-f007:**
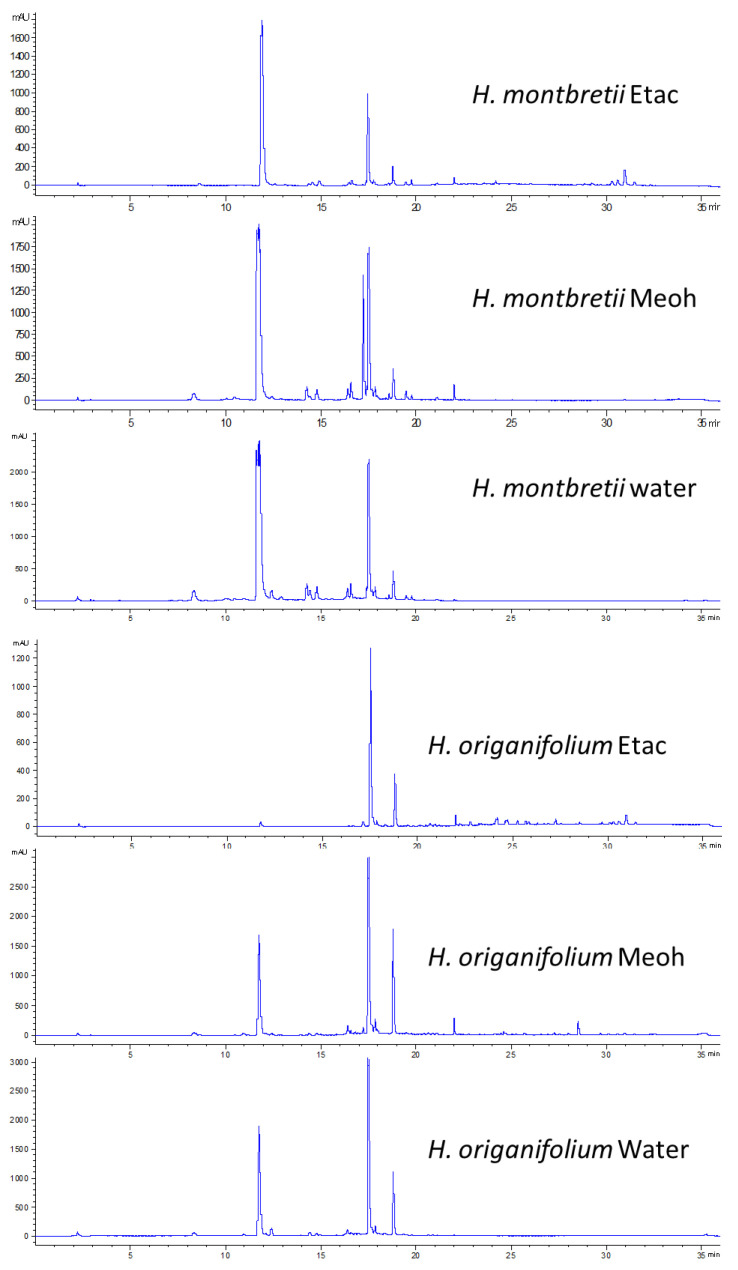
Exemplificative chromatograms at 330 nm of *H. montbretii* and *H. origanifolium* extracts.

**Figure 8 plants-12-00648-f008:**
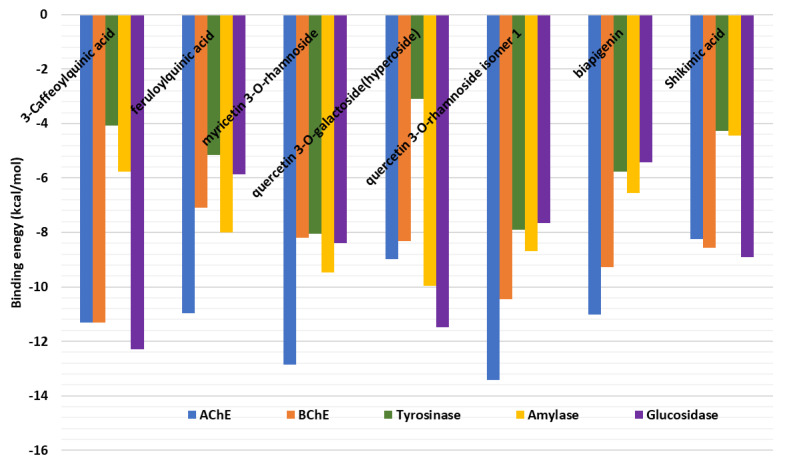
Binding energy (docking) scores of the bioactive compounds from *Hypericum* extracts.

**Figure 9 plants-12-00648-f009:**
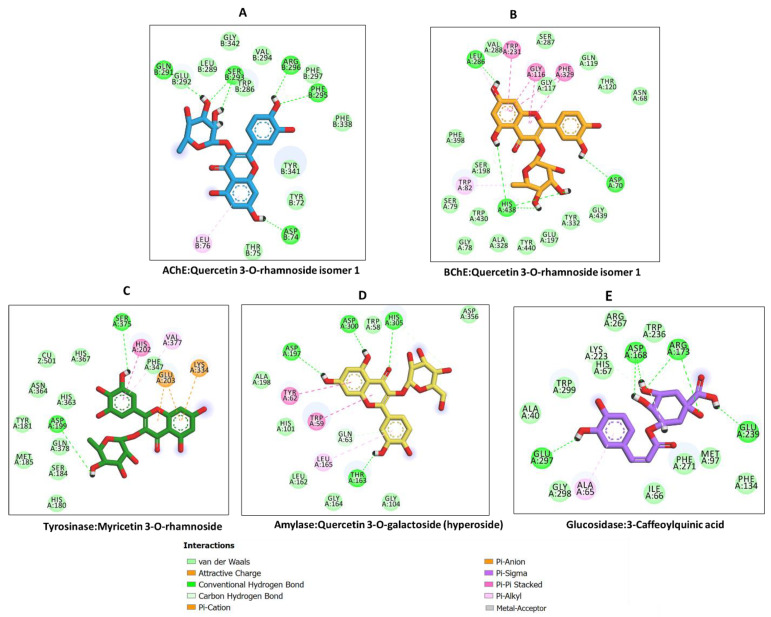
Protein-ligand interaction: (**A**) AChE and quercetin 3-*O*-rhamnoside isomer 1, (**B**) BChE and 3-*O*-rhamnoside isomer 1, (**C**) tyrosinase and myricetin 3-*O*-rhamnoside, (**D**) amylase and quercetin 3-*O*-galactoside (hyperoside) and (**E**) glucosidase and 3-caffeoylquinic acid.

**Figure 10 plants-12-00648-f010:**
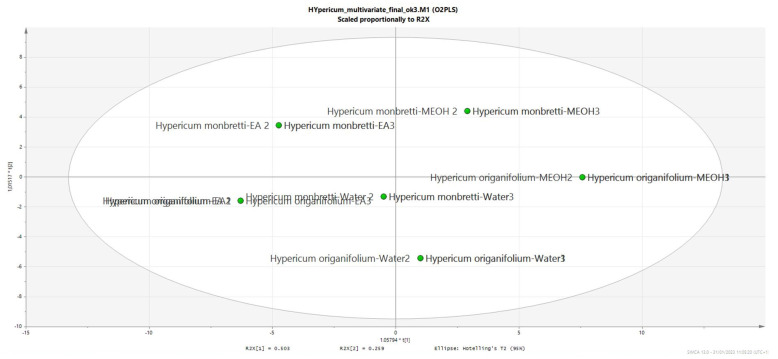
PLS-DA obtained using the matrix obtained with all the quantitative data for the different solvents of each *Hypericum* specie and the results of bioassays. Three replicate were used for generating the plot. Solvents are indicated as EA ethyl acetate, MEOH methanol, Water, number indicate the replicates.

**Figure 11 plants-12-00648-f011:**
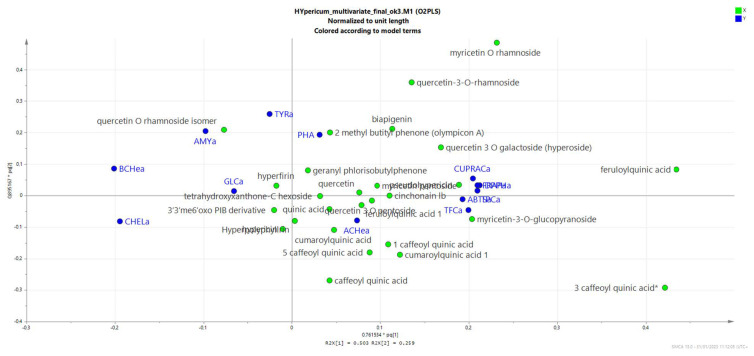
Loading scatter plot corresponding to plot 8, showing the correlation between the compounds that have been detected in the extracts (green dots) and the considered bioassays. Blue dots represent Acetyl Cholinesterase AChea, Butyril Cholinesterase BuCea, Glucosidase GLCa, Tyrosinase TYRa, Amylase AMYa, Total phenolic contents TPC, CUPRAC, FRAP, DPPH, ABTS and Phosphomolibdenum PHA.

**Figure 12 plants-12-00648-f012:**
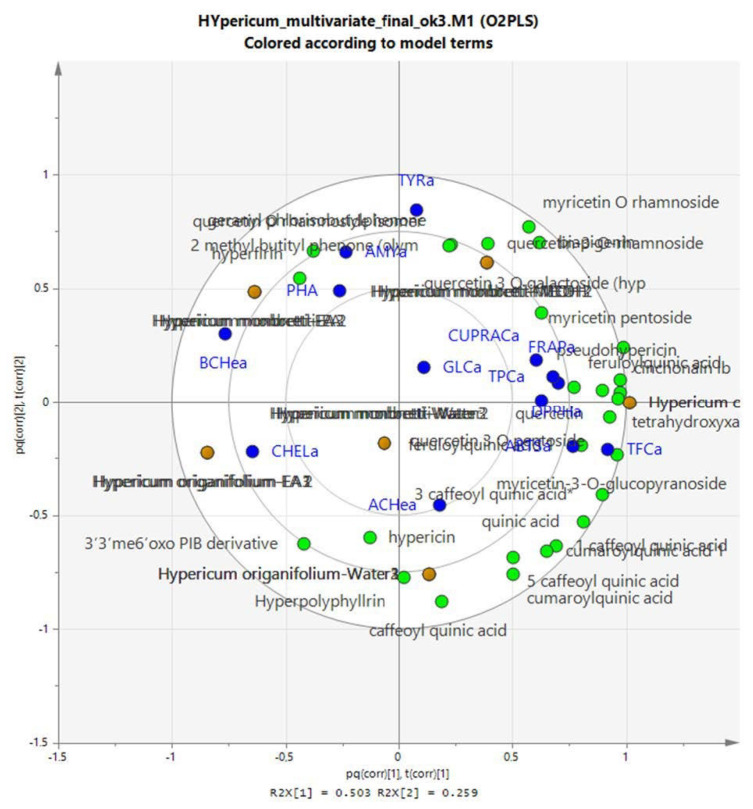
Biplot showing the loading scatter plot of the model generated using the quantitative data on the chemical composition of the extracts and the results of bioassays. Plant extracts are represented with blue squares, enzyme inhibitory test are represented with red dots, metal chelating and ferric reducing power are represented with brown dots, antioxidant assay results are represented with turquoise dots and compounds are all represented with green dots.

**Table 1 plants-12-00648-t001:** Extraction yields (%), total phenolic and flavonoid contents of tested extracts *.

Species	Extracts	Extraction Yields (%)	Total Phenolic Content (mg GAE/g)	Total Flavonoid Content (mg RE/g)
*H. montbretii*	Ethyl acetate	3 ± 0.4	50.97 ± 0.68 ^f^	9.87 ± 0.32 ^f^
	Methanol	23 ± 3	131.11 ± 0.43 ^b^	68.57 ± 0.35 ^b^
	Water	19 ± 2	134.99 ± 0.08 ^a^	62.30 ± 0.27 ^d^
*H. origanifolium*	Ethyl acetate	4 ± 0.5	71.58 ± 0.29 ^e^	25.38 ± 0.27 ^e^
	Methanol	15 ± 2	106.92 ± 2.00 ^c^	82.63 ± 0.47 ^a^
	Water	12 ± 2	93.34 ± 0.38 ^d^	63.59 ± 0.79 ^c^

* Values are reported as mean ± SD. GAE: Gallic acid equivalent; RE: Rutin equivalent. Different letters indicate significant differences between the tested extracts (*p* < 0.05).

**Table 2 plants-12-00648-t002:** Principal NMR assignments obtained from the offline NMR measurements of the extracts *Hypericum* species.

δH (ppm)	δC (ppm)	HMBC	COSY/TOCSY	Assignment
0.85 t	18.96	21.7 32.1 46.9	1.29 1.59 2.34	Fatty acid terminal methyl groups
0.96 d J = 7.5	16.38		3.53	Rhamnose C6
1.29 m	29.34	32.1	2.34	CH2 of fatty acids
1.59 m	24.58		1.29 2.34	CH2 of fatty acids
2.34 m	34.18		1.59	CH2 of fatty acids
2.71–2.30	29.6	138.3 127.9 71.89		CH2 shikimic acid
3.30 m	70.0	67.8	3.33	CH of sugar unit (C-2 glucose)
3.33 m	71.89	67.8 70.2	3.79	CH of sugar unit (C-2 glucose)
3.52 m	62.43	67.2 71.3	3.33	CH2 of sugar unit
3.53 m	70.75			CH of sugar unit
3.69 m	71.4	138.3 29.6		CH shikimic acid
3.81 m	70.35			CH of sugar unit
3.99 m	67.12	127.9 71.4 70.35		CH shikimic acid
4.09	70.04		2.02	CH of quinic acid moiety
4.14	69.03		2.25	CH of quinic acid moiety
4.18	70.04		2.18	CH of quinic acid moiety
4.23	70.04		2.18	CH of quinic acid moiety
4.39	66.13		3.68	CH of sugar unit
4.90	78.05			CH of sugar unit
5.30 brs	102.08		135.5 70.90	Anomeric proton signal of rhamnose linked to position 3 of flavonol
6.20	98.25	95.2 105.3 163.5		C-6 of flavonol derivatives
6.27 d J = 15	113.8	168.3 127.3		C8 hydroxycinnamic moiety
6.37	93.24	98.2 105.3 156.8 163.5		C-8 flavonol derivatives
6.81	137.4	146.5 126.3 71.2		C-2 shikimic acid
6.94	107.80	107.8 146.3 156.8 135.5		C-2′-6′ myricetin derivatives
7.21	124.09 brd			Aromatic moiety
7.32	128.12 m			Aromatic moiety
7.36	128.71 m			Aromatic moiety
7.44	118.53 m			Aromatic moiety
7.45	124.85			Aromatic moiety
7.46	118.53			Aromatic moiety
7.57 d J = 15	145.6	168.3 127.4 121.2 113.8		C7 hydroxycinnamic moiety

**Table 3 plants-12-00648-t003:** Identification and quantification (µg/mg) of secondary metabolites in *H. montbretii* by LC-DAD-MS^n^.

RT	Compound	M-H	Fragments	Ethyl Acetate	Methanol	Water
	quinic acid derivatives					
2.3	quinic acid	191.0555		n.d.	n.d.	n.d.
8.4	1 caffeoyl quinic acid	353.08722		0.10 ± 0.02	1.51 ± 0.09	2.19 ± 0.09
11.9	3 caffeoyl quinic acid *	353.08724		5.20 ± 0.06	22.83 ± 0.15	16.82 ± 0.11
12.5	5 caffeoyl quinic acid	353.08723		-	1.35 ± 0.08	2.20 ± 0.08
14.6	caffeoyl quinic acid	353.08722		0.51 ± 0.04	0.86 ± 0.05	3.58 ± 0.11
14.8	methylferuloylshikimic acid methyl ester	377.1259		0.66 ± 0.04	2.69 ± 0.11	2.88 ± 0.10
16.9	methylferuloylshikimic acid methyl ester	377.1259		0.20 ± 0.03	0.55 ± 0.03	0.77 ± 0.03
16.9	feruloylquinic acid	367.10291		0.07 ± 0.01	0.61 ± 0.03	0.74 ± 0.03
17.3	feruloylquinic acid	367.10291	191 179 135	0.06 ± 0.01	30.29 ± 0.15	0.16 ± 0.01
	flavonoids					
16.7	myricetin hexoside	479.0827	316 287 271	0.57 ± 0.04	4.54 ± 0.13	3.14 ± 0.10
17.5	myricetin pentoside	449.0723	316	0.28 ± 0.10	1.47 ± 0.08	0.80 ± 0.10
17.7	myricetin-3-*O*-rhamnoside	463.0879	316	6.89 ± 0.09	19.77 ± 0.11	11.47 ± 0.10
18	Quercetin-3-*O*-galactoside(hyperoside)	463.0878	301 271	0.70 ± 0.04	3.60 ± 0.11	3.75 ± 0.09
18.7	quercetin-3-*O*-pentoside	433.0775	301	0.27 ± 0.02	1.27 ± 0.04	1.13 ± 0.04
18.8	cinchonain lb	451.1030	341 323 217	0.20 ± 0.01	1.67 ± 0.09	1.23 ± 0.09
18.9	quercetin rhamnoside	447.0926	301	1.84 ± 0.03	6.70 ± 0.12	4.24 ± 0.11
21.1	quercetin	301.0347	271	0.14 ± 0.02	0.84 ± 0.04	0.35 ± 0.02
22.1	biapigenin	537.0822	443 385	0.56 ± 0.04	3.15 ± 0.07	0.28 ± 0.02
	anthraquinone derivative					
14.3	tetrahydroxyxanthone-C hexoside	421.0770	331 301 273	0.01 ± 0.01	0.14 ± 0.01	0.11 ± 0.01
35	hypericin	503.0767	487 459 433	0.01 ± 0.01	0.01 ± 0.01	0.98 ± 0.01
35.1	pseudohypericin	519.0716	503 487 475	0.01 ± 0.01	3.66 ± 0.05	1.28 ± 0.08
	phloroglucinols					
31.0	3′3′me6′oxo PIB derivative	497.6865	222	0.20 ± 0.01	0.03 ± 0.01	0.13 ± 0.01
31.3	Hyperpolyphyllrin	481.3312	437 233	0.05 ± 0.01	0.01 ± 0.01	0.38 ± 0.01

* Comparison with reference standard.

**Table 4 plants-12-00648-t004:** Identification and quantification (µg/mg) of secondary metabolites in *H. origanifolim* by LC-DAD-MS^n^.

RT	Compound	M-H	Fragments	Ethyl Acetate	Methanol	Water
	quinic acid derivatives					
0.3	quinic acid					
8.4	1 caffeoyl quinic acid	353.08722		0.01 ± 0.01	0.77 ± 0.02	1.14 ± 0.09
11.9	3 caffeoyl quinic acid *	353.08718		0.67 ± 0.02	10.65 ± 0.11	10.19 ± 0.11
12.5	5 caffeoyl quinic acid	353.08721		0.01 ± 0.005	0.42 ± 0.03	2.03 ± 0.09
14.6	caffeoyl quinic acid	353.08720		0.01 ± 0.005	0.14 ± 0.01	1.78 ± 0.08
14.8	cumaroylquinic acid	377.1258		0.01 ± 0.005	0.52 ± 0.02	0.60 ± 0.03
16.9	cumaroylquinic acid	377.1258		0.01 ± 0.005	0.08 ± 0.01	0.13 ± 0.01
16.9	feruloylquinic acid	367.10291		0.05 ± 0.01	0.77 ± 0.01	0.60 ± 0.03
17.3	feruloylquinic acid	367.10291	191 179 135	0.01 ± 0.005	3.00 ± 0.12	0.10 ± 0.03
	flavonoids					
16.7	myricetin hexoside	479.0827	316 287 271	0.32 ± 0.01	2.60 ± 0.11	1.72 ± 0.06
17.7	myricetin-3-*O*-rhamnoside	463.0879	301 271	19.19 ± 0.12	23.15 ± 0.21	13.95 ± 0.12
18	Quercetin-3-*O*-galactoside(hyperoside)	463.0879	301 271	1.85 ± 0.09	7.94 ± 0.09	3.11 ± 0.09
18.9	Quercetin-*O*-rhamnoside isomer 1	447.0926	301	7.83 ± 0.08	13.74 ± 0.21	9.65 ± 0.11
19.1	Quercetin-*O*-rhamnoside isomer 2	447.0926	301	2.95 ± 0.08	1.97 ± 0.06	0.01 ± 0.005
21.1	quercetin	301.0347	271	0.19 ± 0.01	0.40 ± 0.01	0.15 ± 0.01
22.1	biapigenin	537.0822	443 385	1.77 ± 0.08	2.83 ± 0.10	0.31 ± 0.04
	phloroglucinols					
28.2	geranyl phlorisobutylphenone	331.1909	287 262 207	0.08 ± 0.01	0.52 ± 0.01	0.02 ± 0.005
28.6	2 methyl butityl phenone (olympicon A)	345.4531	301 276 261 321	0.46 ± 0.03	3.26 ± 0.09	0.10 ± 0.01
30.5	hyperfirin	467.6606	423 398 329	0.14 ± 0.01	0.01 ± 0.005	0.01 ± 0.005
	anthraquinone derivative					
35	hypericin	503.0767	487 459 433	0.43 ± 0.02	0.01 ± 0.005	0.72 ± 0.03
35.1	pseudohypericin	519.0716	503 487 475	0.16 ± 0.01	2.10 ± 0.09	0.76 ± 0.06

* Comparison with reference standard.

**Table 5 plants-12-00648-t005:** Antioxidant properties of tested extracts *.

Species	Extracts	DPPH (mg TE/g)	ABTS (mg TE/g)	CUPRAC (mg TE/g)	FRAP (mg TE/g)	Phosphomolybdenum (mmol TE/g)	Metal Chelating (mg EDTAE/g)
*H. montbretii*	Ethyl acetate	60.39 ± 0.51 ^e^	63.69 ± 0.69 ^f^	175.88 ± 1.81 ^f^	84.74 ± 0.31 ^f^	1.79 ± 0.04 ^bc^	22.61 ± 0.37 ^b^
	Methanol	346.63 ± 2.86 ^a^	360.22 ± 0.56 ^b^	699.16 ± 2.52 ^a^	371.25 ± 5.23 ^a^	1.95 ± 0.08 ^ab^	8.52 ± 0.57 ^e^
	Water	340.05 ± 4.36 ^a^	329.71 ± 11.45 ^c^	637.33 ± 3.03 ^b^	348.91 ± 5.78 ^b^	2.03 ± 0.12 ^a^	4.41 ± 0.22 ^f^
*H. origanifolium*	Ethyl acetate	96.34 ± 2.28 ^d^	254.13 ± 0.11 ^e^	231.71 ± 7.03 ^e^	105.74 ± 1.34 ^e^	1.98 ± 0.03 ^a^	24.27 ± 0.93 ^a^
	Methanol	266.36 ± 2.75 ^b^	402.73 ± 2.16 ^a^	455.56 ± 9.17 ^c^	264.35 ± 0.77 ^c^	1.75 ± 0.04 ^c^	10.29 ± 0.14 ^d^
	Water	194.43 ± 2.67 ^c^	280.63 ± 4.23 ^d^	364.27 ± 3.67 ^d^	216.18 ± 3.40 ^d^	1.46 ± 0.02 ^d^	20.63 ± 0.34 ^c^

* Values are reported as mean ± SD. TE: Trolox equivalent; EDTAE: EDTA equivalent. Different letters indicate significant differences between the tested extracts (*p* < 0.05).

**Table 6 plants-12-00648-t006:** Enzyme inhibitory effects of the tested extracts *.

Species	Extracts	AChE (mg GALAE/g)	BchE (mg GALAE/g)	Tyrosinase (mg KAE/g)	Amylase (mmol ACAE/g)	Glucosidase (mmol ACAE/g)
*H. montbretii*	Ethyl acetate	na	6.61 ± 0.08 ^a^	68.38 ± 0.29 ^a^	0.61 ± 0.02 ^a^	1.00 ± 0.02 ^c^
	Methanol	2.17 ± 0.12 ^b^	na	68.84 ± 0.62 ^a^	0.52 ± 0.01 ^b^	1.14 ± 0.02 ^a^
	Water	1.55 ± 0.11 ^c^	na	34.47 ± 1.08 ^c^	0.05 ± 0.01 ^c^	na
*H. origanifolium*	Ethyl acetate	3.09 ± 0.22 ^a^	4.94 ± 0.15 ^b^	57.47 ± 0.98 ^b^	0.63 ± 0.01 ^a^	1.06 ± 0.02 ^b^
	Methanol	2.17 ± 0.06 ^b^	0.66 ± 0.06 ^c^	69.66 ± 0.47 ^a^	0.49 ± 0.01 ^b^	1.09 ± 0.03 ^b^
	Water	2.21 ± 0.04 ^b^	na	16.63 ± 1.46 ^d^	0.06 ± 0.01 ^c^	1.10 ± 0.02 ^ab^

* Values are reported as mean ± SD. GALAE: Galantamine equivalent; KAE: Kojic acid equivalent, ACAE: Acarbose equivalent; na: not active. Different letters indicate significant differences between the tested extracts (*p* < 0.05).

## Data Availability

No applicable.

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
