# Peer review of "Novel Signposts on the Road from Natural Sources to Pharmaceutical Applications: A Combinative Approach between LC-DAD-MS and Offline LC-NMR for the Biochemical Characterization of Two *Hypericum* Species *(H. montbretii* and *H. origanifolium*)"

_plants, 2023, doi:10.3390/plants12030648_

Round 1
Reviewer 1 Report
1. It is nesesary to bring at least a few references to publications on the study of lipophilic constituents that contribute to antioxidant activity, because methanol, and even more so, ethyl acetate extract contains a significant percentage of such compounds.
2. The names of the used chemical compounds should be written correctly. For example, galantamine, cholinesterase inhibitor, twice named with errors.
3. The names of the used chemical compounds should be written from lowercase, not capital letters.
4. The names of the species of plant should be written in italics.
5. It is necessary to highlight in a separate part at the end of the text a decoding of abbreviations.
6. It is necessary to indicate the yield of raw extracts with solvents of various polarities so that the reader can appreciate the completeness of extraction.
7. It is necessary to analyze the obtained extracts not only on the wavelength of 330nm, but also on other wavelengths, in particular, at 592 nm, in order to adequately assess the content of hyperitsins in extracts.
8. Data of tables 3 and 4 are doubtful due to the presence of values 0.01± 0.01 in them.
9. There is a review: The Phytochemistry and Pharmacology of Hypericum, Chuan-Yun Xiao, Qing Mu, Simon Gibbons// Prog Chem Org Nat Prod. 2020; 112:85-182. Doi:10.1007/978-3-030-52966-6_2. It should be included in the list of literature, because it describes in detail more than 700 components allocated from various types of St. John's wort (Hipericum).
Author Response
- It is necsesary to bring at least a few references to publications on the study of lipophilic constituents that contribute to antioxidant activity, because methanol, and even more so, ethyl acetate extract contains a significant percentage of such compounds.
Response: Thank you reviewer for your comment. From our measurement we observed in ethyl acetate mainly unsaturated lipids that can be responsible for the antioxidant activity we add this information as requested.
- The names of the used chemical compounds should be written correctly. For example, galantamine, cholinesterase inhibitor, twice named with errors.
Response: We apologies for mistake and we corrected all the typos.
- The names of the used chemical compounds should be written from lowercase, not capital letters.
Response: We apologies for mistake and we corrected all the typos.
- The names of the species of plant should be written in italics.
Response: We apologies for mistake and we corrected all the typos.
- It is necessary to highlight in a separate part at the end of the text a decoding of abbreviations.
Response: We inserted a list of abbreviations in the revised version.
- It is necessary to indicate the yield of raw extracts with solvents of various polarities so that the reader can appreciate the completeness of extraction.
Response: Thank you for this good suggestion we completed as requested. The extraction yields were inserted in Table 1.
- It is necessary to analyze the obtained extracts not only on the wavelength of 330nm, but also on other wavelengths, in particular, at 592 nm, in order to adequately assess the content of hyperitsins in extracts.
Response: Thank you reviewer for your comment, we acquired chromatograms in the range 200-650 so the wavelength 592 was recorded for the analysis of Hypericins
- Data of tables 3 and 4 are doubtful due to the presence of values 0.01± 0.01 in them.
Response: Thank you reviewer for your comment, we implemented the decimals and improved tables
- There is a review: The Phytochemistry and Pharmacology of Hypericum, Chuan-Yun Xiao, Qing Mu, Simon Gibbons// Prog Chem Org Nat Prod.2020; 112:85-182. Doi:10.1007/978-3-030-52966-6_2. It should be included in the list of literature, because it describes in detail more than 700 components allocated from various types of St. John's wort (Hipericum).
Response: Thank you reviewer for your comment, the review is largely informative and important so we included in the introduction.
Reviewer 2 Report
REVIEWER’S COMMENT
Major revision comments
The manuscript is very well prepared and presented. Some minor omissions are noted.
1. Results and Discussion:
Lines 482-499: Regarding the tyrosinase inhibitory activity of all extracts it is not clarified whether this is a beneficial effect of the extracts or not and what is the potential application of these extracts in cosmetics. Considering that tyrosinase is responsible for melanogenesis, it can be assumed that a possible beneficial effect can be expected in cases of hyperpigmentation. This should be explained and supported by literature data.
Line 581: In Figure 9 more detailed legend explaining what each color represents should be included. Also which parameters are plotted on the abscissa and which on the ordinate and the abbreviations used should be specified in the legend of both figure 9 and 10.
The novelty of this study should be clearly explained. The strengths and the limitations of the study should be specified.
Overall Recommendation
The paper can in principle be accepted after minor revision.
Author Response
Referee 2
Major revision comments
The manuscript is very well prepared and presented. Some minor omissions are noted.
- Results and Discussion:
Lines 482-499: Regarding the tyrosinase inhibitory activity of all extracts it is not clarified whether this is a beneficial effect of the extracts or not and what is the potential application of these extracts in cosmetics. Considering that tyrosinase is responsible for melanogenesis, it can be assumed that a possible beneficial effect can be expected in cases of hyperpigmentation. This should be explained and supported by literature data.
Response: We improved the section for tyrosinase inhibitory effect in the revised version.
Line 581: In Figure 9 more detailed legend explaining what each color represents should be included. Also which parameters are plotted on the abscissa and which on the ordinate and the abbreviations used should be specified in the legend of both figure 9 and 10.
Response: We thank for comment we corrected
The novelty of this study should be clearly explained. The strengths and the limitations of the study should be specified.
Response: We thank for comment we corrected
Overall Recommendation
The paper can in principle be accepted after minor revision.